# Construction of Leaderless-Bacteriocin-Producing Bacteriophage Targeting *E. coli* and Neighboring Gram-Positive Pathogens

Yoshimitsu Masuda,[a] Shun Kawabata,[a] Tatsuya Uedoi,[a] Ken-ichi Honjoh,[a] Takahisa Miyamoto[a]

[a]Laboratory of Food Hygienic Chemistry, Department of Food Biotechnology, Faculty of Agriculture, Kyushu University, Fukuoka, Japan

**ABSTRACT** Lytic bacteriophages are expected as effective tools to control infectious bacteria in human and pathogenic or spoilage bacteria in foods. Leaderless bacteriocins (LLBs) are simple bacteriocins produced by Gram-positive bacteria. LLBs do not possess an N-terminal leader peptide in the precursor, which means that they are active immediately after translation. In this study, we constructed a novel antimicrobial agent, an LLB-producing phage (LLB-phage), by genetic engineering to introduce the LLB structural gene into the lytic phage genome. To this end, *lnqQ* (structure gene of an LLB, lacticin Q) and *trxA*, an essential gene for T7 phage genome replication, were integrated in tandem into T7 phage genome using homologous recombination in *Escherichia coli* host strain. The recombinant *lnqQ*-T7 phage was isolated by a screening method using Δ*trxA* host strain. *lnqQ*-T7 phage formed a clear halo in agar plates containing both *E. coli* and lacticin Q-susceptible *Bacillus coagulans*, indicating that *lnqQ*-T7 phage could produce a significant amount of lacticin Q. Lacticin Q production did not exert a significant effect on the lytic cycle of T7 phage. In fact, the production of lacticin Q enhanced T7 phage lytic activity and helped to prevent the emergence of bacterial populations resistant against this phage. These results serve as a proof of principle for LLB-phages. There are different types of LLBs and phages, meaning that in the future, it may be possible to produce any number of LLB-phages which can be designed to efficiently control different types of bacterial contamination in different settings.

**IMPORTANCE** We demonstrated that we could combine LLB and phage to construct promising novel antimicrobial agents, LLB-phage. The first LLB-phage, *lnqQ*-T7 phage, can control the growth of both the Gram-negative host strain and neighboring Gram-positive bacteria while preventing the emergence of phage resistance in the host strain. There are several different types of LLBs and phages, suggesting that we may be able to design a battery of LLB-phages by selecting novel combinations of LLBs and phages. These constructs could be tailored to control various bacterial contaminations and infectious diseases.

**KEYWORDS** antimicrobial agents, bacteriocins, bacteriophage genetics, bacteriophages, leaderless bacteriocins

Antimicrobial resistance (AMR) is one of the most significant global health issues of our time. Recently, Tsuzuki et al. reported that over 8,000 deaths could be attributed to bloodstream infections (BSIs) caused by methicillin-resistant *Staphylococcus aureus* (MRSA) and fluoroquinolone-resistant *Escherichia coli* (FQRE) in Japan in 2017 (1). Moreover, the Interagency Coordination Group (IACG) on Antimicrobial Resistance reported to the United Nations that AMR pathogens already cause at least 700,000 deaths globally per year. They also reported that around 2.4 million people could die in high-income countries between 2015 and 2050 without a sustained effort to contain

Address correspondence to Yoshimitsu Masuda, y.masuda@agr.kyushu-u.ac.jp.

antimicrobial resistance (2). This means that it is critical to develop alternative antimicrobial agents to combat AMR pathogens in the future.

Bacteriophages or phages are viruses that infect only bacterial hosts but possess no danger to animal or plant cells. Their replication and propagation are totally dependent on the biosynthetic systems of their host strains. Given their high killing activity against host strains, including AMR pathogens, via bacterial lysis as a result of their propagation, lytic phages are expected to be an effective tool in controlling bacterial infections in the human body and in the decontamination of food products (3). Due to the high specificity of the host strain, phages which infect pathogenic hosts hardly damage any microbial partners within the niche being treated. This means that phages are obvious candidates for developing novel technologies to control AMR pathogens, although there are still some significant challenges in their practical application, including rapid development of resistance in the host strains and instability in environmentally relevant conditions (4–6).

Bacteriocins are ribosomally synthesized antimicrobial peptides produced by bacteria, and bacteriocins produced by Gram-positive bacteria, including lactic acid bacteria (LAB), exhibit bactericidal or bacteriostatic effects against various Gram-positive bacteria, including AMR pathogens, but not against Gram-negative bacteria (7). LAB bacteriocins are detailed as potential safe natural biopreservatives and therapeutic compounds (7–9). The bacteriocins produced by Gram-positive bacteria are classified into two major classes, the so-called lantibiotics and nonlantibiotics, based on the classification described by Cotter et al. (7). Most bacteriocins, including lantibiotics, circular bacteriocins, and most nonmodified bacteriocins, are synthesized as precursor peptides with N-terminal leader peptides that function as recognition sites for posttranslational processes such as secretion by dedicated transporters and modifications for abnormal amino acid residues in the lantibiotics. Leader peptides also function as caps to inactivate bacteriocins inside the producer cells. However, some bacteriocins have been reported to be synthesized without any N-terminal extensions and are termed leaderless bacteriocins (LLBs) (10, 11). Since LLBs have no leader peptides in their structures, they can easily be produced in large quantities by other bacteria or fungi using a simple fusion of their structural genes to producer-specific secretion signals (12–14). Moreover, without N-terminal extension in their structural genes, they are immediately active inside heterologous expression hosts after translation. Lacticin Q (LnqQ) produced by *Lactococcus lactis* QU5 is the most widely studied LLB, possesses high stability against heat and acidic stresses, and exerts strong antibacterial activity against a broad range of Gram-positive bacteria, including pathogenic bacteria such as *Bacillus cereus* and *Staphylococcus aureus* (15–19). Iwatani et al. reported that when only the structural gene of LnaQ (*lnqQ*) was heterologously expressed in the nonproducer strain, the viability of the expressing strain immediately decreased and active LnqQ was detected not from its culture supernatant but from cell fraction (20). Five genes, *lnqBCDEF*, are required for secretion and self-immunity against LnqQ. Liposomal study by Yoneyama et al. demonstrated that LnqQ induced high membrane permeabilization in the absence of specific docking molecule on negatively charged liposome (21), and this mode of action was proposed as the "Huge Toroidal Pore" model (22). These previous studies suggest that heterologously expressed LnqQ in *E. coli* can cause severe damage on the *E. coli* cell membrane.

This study was designed to construct a leaderless-bacteriocin-producing phage (LLB-phage) by introducing a structural gene of LLB into the phage genome. In this study, T7 phage, one of the model phages, was selected as the candidate. The genomic DNA replication of T7 phage requires thioredoxin 1 (trx) (23), and the relatively simple strategy of gene mutation in T7 phage genome using trx gene (*trxA*) deleted mutant was already demonstrated (24, 25). LnqQ-producing T7 phage was the first proof of principle for LLB-phages, and it could lyse its host, *E. coli*, via the lytic process in T7 phage infection and simultaneously kill the surrounding Gram-positive bacteria by releasing LnqQ produced in the infected host, producing a "two birds with one stone" type of strategy.

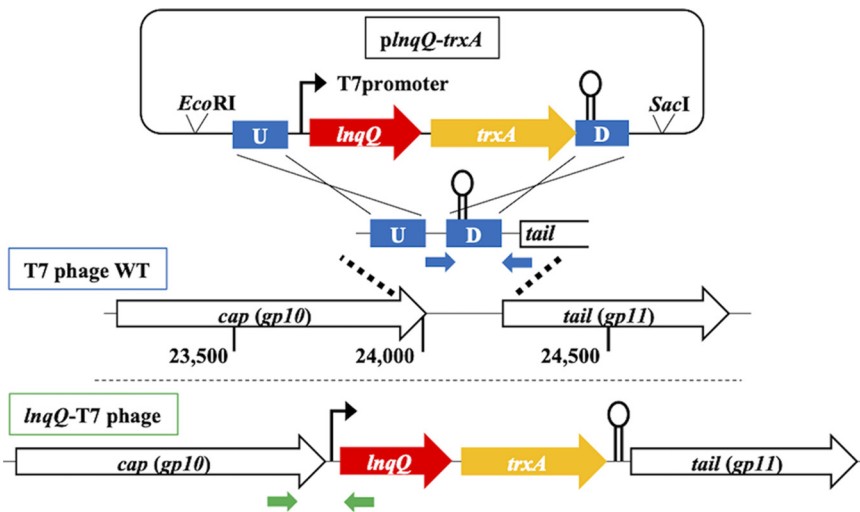

**FIG 1** Scheme of homologous recombination and design of the insertion fragment. The inset where both *lnqQ* and *trxA* were regulated under T7 promoter is integrated between *cap* and *tail* genes in the T7 phage genome by spontaneous homologous recombination in *E. coli* host strain. Upstream and downstream homologous regions are indicated as blue squares labeled U and D, respectively. Two pairs of small blue and green arrows indicate the positions of the primers used for the PCR analyses.

## RESULTS

**Construction and screening of the *lnqQ*-T7 phage.** The construction of lacticin Q-producing T7 phage was performed referring to previous studies by Qimron et al. (24, 25). The LnqQ structural gene (*lnqQ*) was introduced into the T7 phage genome in tandem with *trxA*, which encodes an essential cofactor for the T7 phage DNA replisome (23). The insertion fragment, which incorporated *lnqQ* and *trxA* between two homologous sequences, was designed and cloned into a pETUK vector and produced p*lnqQ-trxA* for homologous recombination. The insertion site of *lnqQ* and *trxA* was determined at the position between *cap*, the gene encoding the capsid protein, and its downstream T7 terminator sequence, as shown in Fig. 1. The gene expression system of T7 phage is basically based on the gene location where all genes are classified into 3 parts, early, middle, and late stages in the infectious cycle (26), and the *lnqQ-trxA* insertion site is located at the late stage while T7 DNA polymerase locates at the middle stage. To ensure enough *trxA* expression for DNA replication at proper timing during the infectious cycle, we set the additional T7 promoter sequence upstream of *lnqQ* and *trxA*.

The phage solution containing the *lnqQ*-T7 recombinant phage was obtained following overnight infection of T7 phage to *E. coli* DH5α harboring p*lnqQ-trxA*. This solution was then utilized in the plaque-forming assay with *E. coli* Δ*trxA* strain used as a host strain to screen out nonrecombinant T7 phage particles (Fig. 2A). T7 phage could not provide any plaques on the Δ*trxA* plate (0 PFU/ml) (Fig. 2B), while the recombinant phage solution provided several clear plaques. The homologous recombination rate was calculated to be ~1/9,000 following comparison of the plaque-forming activity of the recombinant phage solution when using WT and Δ*trxA* strains as hosts. Plaque PCR analysis with those clear plaques on the Δ*trxA* plates demonstrated that these clear plaques were produced largely by the *lnqQ*-T7 phage alone and that relatively few of these plaques were contaminated with nonrecombinant T7 phage (Fig. 2C).

**Double-indicator plaque assays using the *lnqQ*-T7 phage.** A single clear plaque without T7 phage contamination, as confirmed by plaque PCR analysis, was selected, and then the *lnqQ*-T7 phage was purified and propagated by overnight infection to the Δ*trxA* strain. This culture filtrate was subjected to PCR analysis to confirm that there was no contamination with T7 phage and then utilized to determine the titer and plaque-forming activity against both *E. coli* hosts and LnqQ susceptible *Bacillus coagulans*.

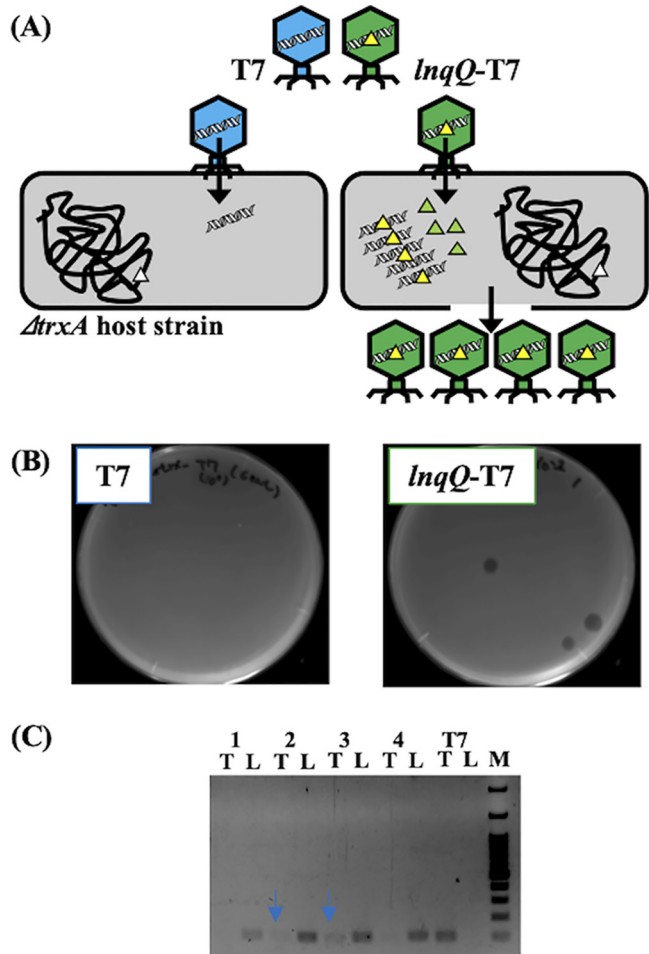

**FIG 2** Screening of the recombinant *lnqQ*-T7 phage with ΔtrxA host strain. (A) Illustration of the screening process of the *lnqQ*-T7 phage from phage solution after homologous recombination. Due to lack of TrxA (green triangle) in ΔtrxA host strain (open triangle indicates lack of *trxA*), only the *lnqQ*-T7 phage which possesses *trxA* in its genome (yellow triangle in phage genome) can replicate its phage particles and lyse the host strain to form clear plaques. (B) Plaque-forming activities of T7 and the *lnqQ*-T7 phages on the ΔtrxA plates after overnight incubation at 37°C. (C) Plaque PCR analysis to confirm insert containing *lnqQ*. Aliquots of the *lnqQ*-T7 phage solutions prepared from clear plaques on the ΔtrxA plates (samples 1 to 4) were directly utilized as templates. T7 phage solution was used as a negative control. T and L indicate the PCR products using the primer pair for T7 and *lnqQ*-T7 phage, respectively. Blue arrows indicate the T7 phage contamination in samples 2 and 3. M, ExcelBand 100 bp+3K DNA Ladder (SMOBIO, Hsinchu City, Taiwan).

The phage was incubated with its host *E. coli* strain in LB medium for 1 h before plating with top agar containing *B. coagulans*, which allowed us to assay the LnqQ production in the host strain during lysis via its impact on the growth of *B. coagulans*. The parental T7 phage formed clear plaques only against *E. coli* wild type (WT) (Fig. 3). When *B. coagulans* was contained in top agar, the growth of *B. coagulans* was shown inside the plaques of T7 phage formed against *E. coli* host (Fig. 3). On the other hand, in addition to the obvious lytic activity against both *E. coli* WT and *E. coli* ΔtrxA host strains, the *lnqQ*-T7 phage also formed clear plaques even in the presence of *B. coagulans* in the top agar, suggesting that LnqQ production was sufficient to kill the surrounding *B. coagulans* cells during the *lnqQ*-T7 phage lysis cycle (Fig. 3).

The antimicrobial activity of the *lnqQ*-T7 phage solution was also evaluated using the spot-on-lawn method. As expected, the parental T7 phage showed antimicrobial activity against only *E. coli* WT but not against ΔtrxA, while the *lnqQ*-T7 phage exhibited clear inhibition zones against both WT and ΔtrxA hosts (Fig. 4A). After overnight incubation, phage-spotted *E. coli* plates with or without clear halos were overlaid with a

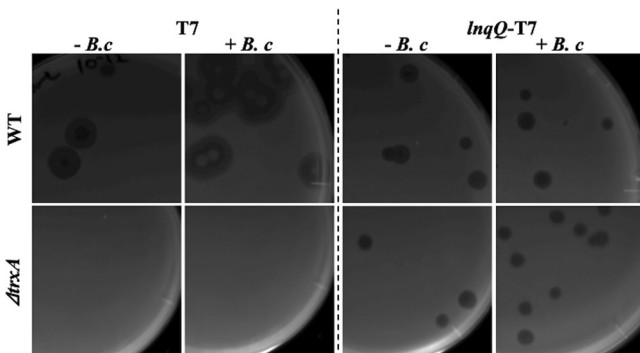

**FIG 3** Plaque-forming assays with T7 and the *lnqQ*-T7 phage. *E. coli* BW25113 (WT) and its *trxA* deletion mutant (Δ*trxA*) were used as the host strains. A total of 100 μl of serially diluted phage solution was mixed with 100 μl of the overnight grown host *E. coli* culture and incubated at 37°C for 1 h. Then, the mixed solution was added to the 5 ml of melted top agar with (+) or without (−) 100 μl of the overnight grown *B. coagulans* NBRC 12714 culture and poured on top of the bottom agar layer (TSA). Dried plates were incubated at 37°C overnight and plaque-forming activities were evaluated.

second top agar containing *B. coagulans* and subsequently incubated overnight to evaluate the LnqQ activity following *lnqQ*-T7 phage-mediated lysis of the *E. coli* hosts. Only the *lnqQ*-T7 phage exhibited antimicrobial activity against *B. coagulans* in the area corresponding to the halos formed on the *E. coli* plates (Fig. 4A). These results strongly suggest that the *lnqQ*-T7 phage can produce sufficient LnqQ to exhibit some antimicrobial activity against surrounding bacteria. The *lnqQ*-T7 phage did not produce a clear halo when spotted on top agar impregnated with both *E. coli* and *B. coagulans* (data not shown). The *lnqQ*-T7 phage needed to preinfect the *E. coli* hosts either in liquid medium or on agar plates before mixing with *B. coagulans* to exhibit any obvious antimicrobial effect via the production of LnqQ. In addition, at least $10^2$ PFU/ml *lnqQ*-T7 phage was required to produce enough LnqQ to inhibit overlaid *B. coagulans* when *E. coli* WT was used as the host strain. When Δ*trxA* was used as the host strain, more than $10^3$ PFU/ml of *lnqQ*-T7 phage was required to demonstrate any antimicrobial activity against *B. coagulans*, even though the efficiency of plaquing (EOP) in the *lnqQ*-T7 phage was almost the same in both host strains (Fig. 4B).

**Latent period and burst size of the *lnqQ*-T7 phage.** Since heterologous expression of *lnqQ* from T7 phage genome could affect the lytic cycle of T7 phage, including changing both the latent time and burst size, one-step growth curve experiments

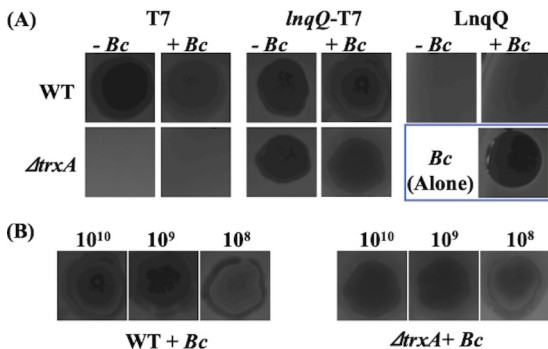

**FIG 4** Antimicrobial activity assays using the spot-on-lawn method. (A) *E. coli* BW25113 (WT) and Δ*trxA* strain were used as the host strains, and 5 μl of phage solutions ($10^{10}$ PFU/ml) were spotted on the top agar plates containing host strains. Then, the dried plates were incubated at 37°C for 24 h (−*Bc*). After 24 h of incubation, these plates were overlaid by the second top agar containing LnqQ-susceptible *B. coagulans* NBRC12714, and then dried plates were incubated at 37°C overnight again (+*Bc*). Purified LnqQ (5 μg/ml) was used as the control and indicated clear halo against a plate containing only *B. coagulans* [*Bc* (Alone)]. (B) Serially diluted *lnqQ*-T7 phage solutions ($10^8$ to $10^{10}$ PFU/ml, 5 μl) were spotted on the *E. coli* WT and Δ*trxA* plates, and then dried plates were utilized for overlay antimicrobial activity assays as described above.

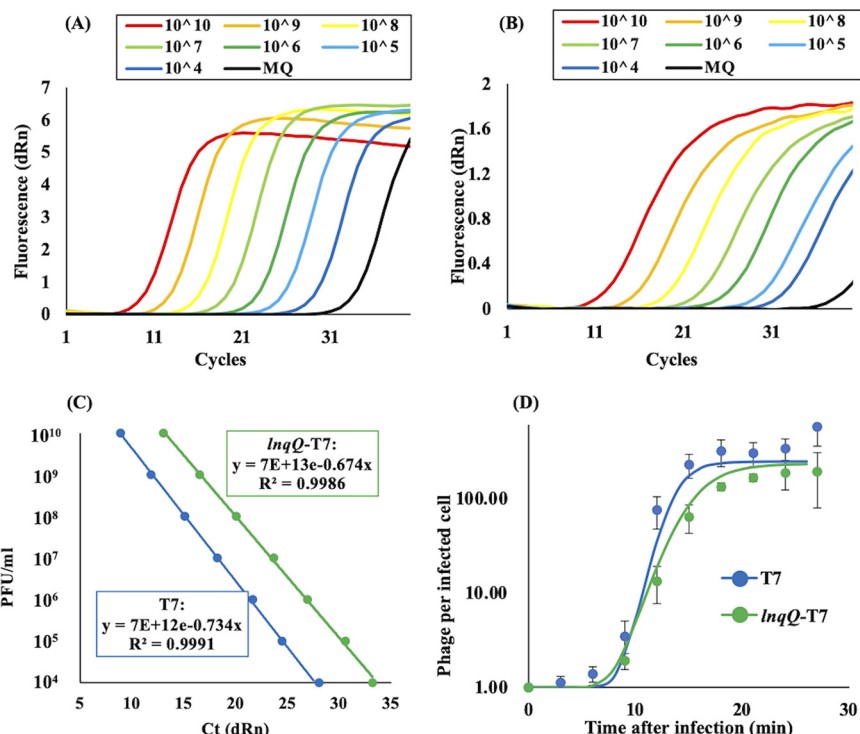

**FIG 5** Phage quantification by qPCR and the one-step growth curves of T7 and the *InqQ*-T7 phages. Amplification plots of T7 (A) and the *InqQ*-T7 (B) phages were obtained by qPCR using Mx300P real-time PCR system and Thunderbird SYBR qPCR master mix according to the manufacturer's instructions. One microliter of each standard phage solution was utilized as a template in each reaction tube (10 μl). Distilled water (MQ, black line) was used as a negative control. (C) Standard curves for both T7 and the *InqQ*-T7 phages were constructed from the results of their amplification plots. (D) One-step growth curves of T7 and the *InqQ*-T7 phages were obtained using the qPCR quantification method. Briefly, exponentially growing host strain *E. coli* BW25113 and the phage solution were mixed in 500 μl of LB at an MOI of 0.01 and incubated at 37°C for 5 min. Infected cells were washed and resuspended in prewarmed 10 ml of LB and incubated at 37°C for 30 min. Cell-free supernatant was transferred into a 1.5-ml tube every 3 min during infection, and the cell-free supernatant was used as a template for the phage quantification qPCR. The burst size was calculated by sigmoidal curve constructed with Fiji, an open-source platform for biological-image analysis using data from three independent infection experiments. The $R^2$ values of sigmoidal curves for both T7 and the *InqQ*-T7 phage one-step growth curves were 0.99998 and 0.99995, respectively.

using both the T7 and *InqQ*-T7 phages were caried out to evaluate these parameters in response to *InqQ* expression. Here, we quantified phage particles not only by using plaque-forming assay but also by qPCR using phage solution as the template, as previously reported by Peng et al. (27). Standard curves for both T7 and *InqQ*-T7 phages were obtained with a significant degree of accuracy ($R^2 > 0.999$), allowing us to calculate the number of phage particles generated in each culture every 3 min over the infection period (Fig. 5A to C). The sigmoidal curves were constructed with Fiji, an open-source platform for biological-image analysis (28), using data from three independent experiments. The $R^2$ values of sigmoidal curves for both T7 and the *InqQ*-T7 phage one-step growth curves were 0.99998 and 0.99995, respectively. Based on the constructed sigmoidal curve, the latent times of both phages were not significantly different, but the total time of their infectious cycles differed. T7 phage reached the stable phase at 18 min, while the *InqQ*-T7 took 24 min to complete a single infectious cycle (Fig. 5D). According to their infectious cycles and qPCR results, the burst sizes of T7 and the *InqQ*-T7 phages were determined as $316 \pm 99$ (at 18 min) and $187 \pm 65$ (at 24 min) phage particles per infected cell of *E. coli* BW25113, respectively, and the significance of the difference between two phages was evaluated by *t* test ($P = 0.1329$). These results indicate that *InqQ* expression slowed down the replication speed of T7 phage and might be able to reduce the burst size of T7 phage.

Microbiology
Spectrum

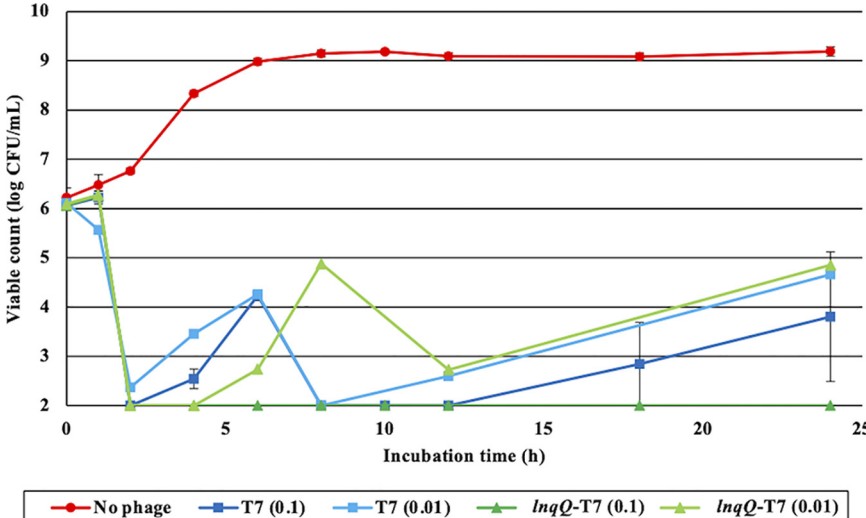

**FIG 6** Effects of T7 and the *lnqQ-T7* phages on the viability of *E. coli* BW25113. *E. coli* BW25113 was inoculated into 10 ml of LB containing T7 phage (MOI of 0.1 [blue line], 0.01 [light blue line]) or *lnqQ-T7* phage (MOI of 0.1 [green line], 0.01 [light green line]). Mixed solutions were incubated at 37°C for 24 h with shaking, and viable cells were counted by CFU on TSA plates at each time point. Bacterial suspension without a phage solution was used as a negative control (red line). The experiments were triplicated, and error bars indicate standard errors of the mean.

**Lytic activity of T7 and *lnqQ-T7* phages.** Since LLB is immediately active inside bacterial cells following translation, LnqQ production could confer not only additional killing activity against Gram-positive neighbors but also enhanced lytic activity in its host strain. Therefore, viable counts of the *E. coli* host strain during phage treatment were monitored to compare the lytic activities of T7 and *lnqQ-T7* phages. There were no differences in the reduction rates of viable counts in the *E. coli* host following infection with either phage, and the viable counts decreased to less than the lower limit of detection ($10^2$ CFU/ml) after 2 h in both cases (Fig. 6). Resistant bacterial populations emerged 4 h after initial T7 phage treatment at MOIs of 0.1 and 0.01, while the *lnqQ-T7* phage, at an MOI of 0.01, retarded the emergence of these resistant populations by at least 2 h (from 6 h). In addition, when the *lnqQ-T7* phage was applied at an MOI of 0.1, no resistant populations were detected over the entire course of incubation (from 2 to 24 h).

## DISCUSSION

Even with the great antimicrobial potential of phages, some significant concerns around the use of phages in clinical and food industry applications remain. For instance, some phages possess several virulence genes in their genome, and lysogenic phages could even assist the horizontal gene transfer of virulence factors among pathogenic bacteria. In addition, the specificity of phages can act as a double-edged sword. Phages can target bacterial strains "precisely" without unwanted effects on the beneficial microbes within the niche, but this also means that a single phage cannot combat all strains of potentially pathogenic bacteria even within the same species, meaning that it is necessary to use phage cocktails with broader specificity (5, 29, 30). Some of these concerns can be mitigated by the application of genetic engineering technologies which can be used to remove potentially harmful genes or improve specificity of host strains, expanding their safe and practical use (31–33). One of the most well-established and widely used methods for phage genome engineering is homologous recombination using phage-host strains. Also, the key point in phage editing is the screening and isolation of modified phage particles from a mixed phage population. So far, methods using clustered regularly interspaced short palindromic repeats (CRISPR) and CRISPR-associated (Cas) genes such as CRISPR-Cas 3, 9, and 10 systems

(31, 34, 35) have shown some promise as editing and screening tools, but they will need to continue to develop before being widely applicable.

Here, we used spontaneous homologous recombination in the host strain and a screening method using a *trxA*-dependent screening process (24, 25) to select our recombinant *trxA*-harboring T7 phage constructs producing LnqQ (Fig. 1 and Fig. 2A). As expected, only the *lnqQ-trxA* integrated phage (*lnqQ*-T7 phage) could form plaques in the Δ*trxA* plates, and *lnqQ-trxA* insertion was confirmed by PCR and DNA sequencing (Fig. 2B and C). LnqQ production from the *lnqQ*-T7 phage was clearly demonstrated as antimicrobial activity against LnqQ-susceptible indicators in the double-indicator plaque-forming and spot-on-lawn assays (Fig. 3 and 4). However, LnqQ itself could not be detected from *lnqQ*-T7 phage solution ($>10^{10}$ PFU/ml) using the LnqQ-optimized LC-MS analysis shown previously by Zendo et al. (36) (data not shown). These results suggest two possibilities. First, because of the attachment of LnqQ to the negatively charged cell membrane, translated LnqQ may be stuck inside the host cells or form a complex with the host cell membranes, meaning that LnqQ was not easily released from the host cells, and only a small amount of LnqQ actually reaches the susceptible strains in close proximity to the host cells. Or, second, the produced LnqQ could simply be too low to be detected. At any rate, the expression level of LnqQ will be optimized in future studies by evaluating the position or increasing the number of the *lnqQ* insertion into the T7 phage genome.

LnqQ productivity differed between the two host strains, *E. coli* BW25113 and its Δ*trxA* mutant. According to the results shown in Fig. 4B, LnqQ production in the WT strain was approximately 20 times higher than that in Δ*trxA* as host strain. Thioredoxin is a ubiquitous protein involved in the various redox reactions associated with a wide range of cellular events, including protein folding, regulation of oxidative stress, and signaling (37). Therefore, reduced protein synthesis in Δ*trxA* mutant may be one of the reasons for its reduced LnqQ production compared to that in WT.

Insertion of the *lnqQ* gene affected the lytic cycle of T7 phage slightly (Fig. 5D). The latent times were not significantly different between T7 and the *lnqQ*-T7 phages, while the burst size of the *lnqQ*-T7 phage ($187 \pm 65$) seemed to be smaller than that of parental T7 phage ($316 \pm 99$). In addition, according to the constructed sigmoidal curves, the total time to complete a single infectious cycle of the *lnqQ*-T7 phage (24 min) was longer than that of the T7 phage (18 min). These results are consistent with the characteristics of LnqQ as an LLB. LnqQ could be produced during the lytic cycle and stored within the host cells, making it immediately active and able to attack the cell membrane from the inside, which might result in the earlier cell lysis (shorter latent time) and the reduced phage particle number (smaller burst size and slower growth) associated with the *lnqQ*-T7 phage. LnqQ production, however, did not exert a significant effect on the lytic cycle of T7 phage. In fact, LnqQ production enhanced T7 phage lytic activity and helped to prevent the emergence of bacterial populations resistant against this phage (Fig. 6). Most of LAB bacteriocins cannot attack the Gram-negative bacteria because of their complex outer membrane system, but once the outer membrane is damaged by metal chelators such as EDTA, citric acid, or lactic acid, the LLB bacteriocins can interact with the target cytoplasmic cell membrane and inhibit the growth of Gram-negative bacteria (38–40). LnqQ produced by the *lnqQ*-T7 phage in host cells can interact directly with the cell membrane of the host cell, which may damage cellular functions and impair cell growth. This disordered membrane function seems to cause severe reductions in several cellular activities, including the expression of the phage-resistance systems such as the CRISPR-Cas system, explaining the reduction in resistant bacteria following the *lnqQ*-T7 phage infection.

In conclusion, we were able to successfully combine LLB and phage using a genetic engineering approach, and the LLB-phage construct was able to exert some antimicrobial activities on both the host cells and surrounding Gram-positive bacteria. Moreover, our LLB-phage could also slow down development of resistance to the phage in the host strain, potentially extending its efficacy. This is just the beginning of the evaluations of LLB-phages, and several challenges remain. These include the construction of more versatile phage genome editing systems, such as CRISPR-Cas systems, which will facilitate the

utilization of different types of phages in different host strains. Low productivity of LLB should be optimized, for example, by reevaluating the location of LLB structure gene or increasing the insertion of structure gene number. Despite this, our results serve as a proof of principle. There are different types of LLBs, such as aureocin A53 (41, 42) and weissellicin YM (43), and mature peptides from other nonmodified bacteriocins can also be the candidates for LLB-phage design. In addition to many choices of LLBs, there are an almost countless number of phages, meaning that in the future, it may be possible to produce any number of LLB-phages which can be designed to efficiently control different types of bacterial contamination in different settings.

## MATERIALS AND METHODS

**Bacterial strains, phages, and media.** *Escherichia coli* BW 25113 and Δ*trxA* were obtained from the National BioResource Project (NBRP), Shizuoka, Japan. *Escherichia coli* DH5α was purchased from TaKaRa Bio (Shiga, Japan), and *Bacillus coagulans* NBRC 12714 was obtained from National Institute of Technology and Evaluation at the Biological Resource Center (NBRC), Kisarazu, Chiba, Japan. LB medium (Difco) and tryptic soy agar (TSA; Difco) were used for bacterial cultures. All bacterial strains were stored at −80°C in glycerol stocks and then precultured at 37°C overnight in LB medium with shaking from a single colony grown on TSA plates, prior to experimentation. Culture media and agar plates for the recombinant strains were supplemented with 10 μg/ml of kanamycin.

T7 phage was purchased from the NBRC. After overnight infection to exponentially growing *E. coli* DH5α at 37°C in LB medium with shaking, cells were collected and the supernatant was filtered through a 0.45-μm membrane (Merck Millipore, Ireland) to produce the T7 phage solution. This filtered T7 phage solution ($10^{10}$ PFU/ml) was stored at 4°C and used for subsequent experiments.

**Plasmid construction for homologous recombination.** DNA polymerase, restriction enzymes, and other DNA-modifying enzymes were used according to the manufacturer's instructions. The insert DNA fragment containing *lnqQ*, *trxA*, and homologous sequences was synthesized by Genewiz (South Plainfield, NJ) and ligated into vector plasmid pETUK (Biodynamics Laboratory Inc., Japan) using Ligation high Ver.2 (TOYOBO, Japan) after digestion with EcoRI and SacI (NEB, MA) (Fig. 1). The ligation product was introduced into *E. coli* DH5α by electroporation using ECM 630 electroporation generator (BTX, Holliston, MA) according to the procedure recommended in the manufacturer's instructions. The correct plasmid construct, p*lnqQ-trxA*, was extracted from the recombinant strain using FastGene plasmid minikit (NIPPON Genetics, Japan), and the sequence of the insert for homologous recombination was confirmed by DNA sequencing (Genewiz).

**Homologous recombination and screening with *trxA* mutant.** Homologous recombination occurred naturally in *E. coli* DH5α with p*lnqQ-trxA* during the infection cycle of T7 phage. Briefly, T7 phage was mixed with exponentially growing *E. coli* DH5α at a multiplicity of infection (MOI) of 1.0 in 5 ml of LB medium and incubated at 37°C for overnight with shaking. The overnight culture was centrifuged (4°C, 5,000 × *g*, 3 min) and the supernatant was filtered as described above to obtain the recombinant phage solution. To screen single plaques of the recombinant T7 phage containing *lnqQ* and *trxA* in its genomic DNA (*lnqQ*-T7 phage), this filtered phage solution was used in the plaque assay with *E. coli* BW25113 Δ*trxA* as the host strain by reference to previous studies by Qimron et al. (24, 25) and in the plaque PCR analysis to confirm the introduction of *lnqQ* into T7 phage genome.

**Plaque assay.** Plaque assays were performed using double-layer method, as previously described by Son et al. (44) with minor modifications. Briefly, 100 μl of the phage solution was mixed with 100 μl of exponentially growing host culture and incubated at 37°C for 1 h. Then, the phage-host solution was mixed with 5 ml of melted top agar (LB containing 0.6% agarose, 55°C) and poured on top of the bottom agar layer (TSA). After incubation at 37°C overnight, the number of clear plaques was counted to determine the PFU of the phage solutions. *E. coli* BW25113 Δ*trxA* was used as the host strain for the screening of the *lnqQ*-T7 phage.

**Plaque PCR analyses.** The *lnqQ*-T7 phage solutions were prepared from clear plaques on the Δ*trxA* plates. Agar plugs from the clear plaques were excised using micropipette tips and mixed with 500 μl of saline magnesium (SM) buffer (0.05-M Tris-HCl buffer [pH 7.5], containing 0.1-M NaCl, 8-mM MgSO4, and 0.01% gelatin) and then vortexed for 1 min. The crude phage solution was then centrifuged and filtered as described above. Aliquots of the phage solutions were then directly applied as templates in the PCR analysis using Go *Taq* master mix (Promega). We designed a specific set of primers to target the original T7 phage (T7-24041Fw: 5′-GAAGAGG-CGAGTGTTACTTCAACAG-3′/T7-24146Rv: 5′-GCAGC-AGCCAACTC AGCTTC-3′) and the recombinant *lnqQ*-T7 phage (T7-23916Fw: 5′-AAGC-TGCTGGTGCAGTG-3′/lnqQ-Rv: 5′-GGGAAACCGTTGT-GGTCTC-3′), which were then used in the PCR analysis (Fig. 1).

**Purification of the *lnqQ*-T7 phage.** The *lnqQ*-T7 phage in SM buffer was once again mixed with *E. coli* BW25113 Δ*trxA* mutant at an MOI of 1.0 in 5 ml of LB medium and incubated at 37°C overnight with shaking. Overnight culture supernatant was then collected by centrifugation and filtered to obtain a purified *lnqQ*-T7 phage solution ($10^{10}$ PFU/ml). To confirm the absence of the original T7 phage in the *lnqQ*-T7 phage solution, this solution was applied directly as the template in the selective PCR using T7 and *lnqQ*-T7 targeting primers described above.

**Double-indicator plaque assay.** To evaluate the LnqQ production from the *lnqQ*-T7 phage, a double-indicator plaque assay was designed using T7 phage host *E. coli* and the LnqQ-susceptible indicator *Bacillus coagulans* NBRC 12714. Serially diluted phage solutions (100 μl) were first mixed with exponentially growing host cultures of *E. coli* BW25113 WT or Δ*trxA* (>$10^9$ CFU/ml, 100 μl) and incubated at 37°C for 1 h. After incubation, the infection mix and exponentially growing *B. coagulans* culture (>$10^9$ CFU/ml,

60 $\mu$l) were then added to 5 ml of melted top agar and poured onto bottom agar. Plates were then incubated at 37°C overnight and plaque-forming activity was evaluated as the antimicrobial activity of the phage solution against both *E. coli* host strains and LnqQ-susceptible *B. coagulans*. The LnqQ productions in WT and Δ*trxA* strains were compared using the minimum phage concentration which exhibited a clear halo on the overlaid top agar containing *B. coagulans*.

**ESI-LC-MS analysis.** LnqQ in the *lnqQ*-T7 phage solution and infected *E. coli* sediments was partially purified using the previously described acetone precipitation method (20, 36) and used in liquid chromatography-mass spectrometry (LC-MS) analysis to confirm LnqQ production. LC-MS analysis was conducted using the Agilent 1100 high-performance liquid chromatography system (Agilent Technologies, Palo Alto, CA) equipped with a JMS-T100LC (JEOL, Tokyo, Japan) electrospray ionization (ESI) time-of-flight mass spectrometer according to the previously described method (20).

**Antimicrobial activity assays using spot-on-lawn method.** The antimicrobial activity of the *lnqQ*-T7 phage was also evaluated using spot-on-lawn method. A total of 5 $\mu$l of serially diluted phage solutions was spotted on top agar containing *E. coli* host strains (100 $\mu$l of exponentially growing culture in 5 ml of top agar), and plates were dried at room temperature before incubation at 37°C for 24 h. LnqQ production was then evaluated by overlaying the cultured plates with a second top agar containing LnqQ-susceptible *B. coagulans* NBRC12714. These plates were also dried and incubated at 37°C overnight. Purified LnqQ (5 $\mu$g/ml) was kindly provided by Takeshi Zendo at Kyushu University and used as a control.

**One-step growth curve experiments.** One-step growth curve experiments were performed according to the method described by Son et al. (44) with some modifications. Exponentially growing host strain *E. coli* BW25113 and T7 or *lnqQ*-T7 phage solutions were mixed at a final concentration of $10^9$ CFU/ml and $10^7$ PFU/ml in 500 $\mu$l of LB (MOI of 0.01) and incubated at 37°C for 5 min. Mixed solutions were centrifuged (25°C, 12,000 $\times$ g, 3 min), and the supernatant was discarded to remove cell-free phage particles. Pellets were resuspended in 10 ml of prewarmed LB (37°C), and mixed solutions were incubated at 37°C for 30 min. Every 3 min during infection, 200 $\mu$l of solution was transferred into a 1.5-ml Eppendorf tube and centrifuged (4°C, 12,000 $\times$ g, 3 min). After centrifugation, supernatants were transferred into new tubes and kept at 4°C as cell-free phage solutions until they were used as templates for phage quantification by quantitative PCR (qPCR). The burst size of each phage was calculated by dividing the phage particle number at the end of the single infectious cycle by the phage particle number at time zero. The sigmoidal curves were constructed with Fiji, an open-source platform for biological-image analysis (28), using data from three independent experiments. The $R^2$ values of sigmoidal curves for both T7 and *lnqQ*-T7 phage one-step growth curves were 0.99998 and 0.99995, respectively.

**Phage quantification by qPCR.** In this study, the number of phage particles in each phage solution was quantified using not only plaque-forming assay but also qPCR using each phage solution as a template as previously described (27). qPCR was performed using the Mx3000P real-time PCR system (Stratagene, La Jolla, CA, USA) and Thunderbird SYBR qPCR master mix (TOYOBO Co., Ltd., Osaka, Japan) according to the manufacturer's instructions. Phage solutions were first quantified by plaque-forming assay and then adjusted for serial dilution and the preparation of the standard solutions ($10^4$ to $10^{10}$ PFU/ml). Standard curves for both T7 and *lnqQ*-T7 phages were constructed using these adjusted phage solutions as templates (1 $\mu$l/10 $\mu$l of reaction solution) and using the primers T7-24041Fw/T7-24146Rv and T7-23916Fw/lnqQ-Rv.

**Bacterial resistance against T7 and *lnqQ*-T7 phages.** To evaluate the emergence of phage-resistance populations within the host strain, we went on to perform the time course analysis of lytic activities of T7 and *lnqQ*-T7 phages against *E. coli* BW25113. Exponentially growing host strain *E. coli* BW25113 and T7 or *lnqQ*-T7 phage solutions were mixed at a final concentration of $10^6$ CFU/ml and $10^4$ to $10^5$ PFU/ml in 10 ml of LB (MOI of 0.01, 0.1) and incubated at 37°C for 24 h with shaking. At each time point (0, 1, 2, 4, 6, 8, 10, 12, 18, and 24 h), 50 $\mu$l of the cultures was collected and serially diluted in PBS, and then 10 $\mu$l of the diluted cultures was spotted on LB agar plates. After the plates dried, they were incubated at 37°C for 24 h, and viable cell counts were evaluated as CFU per milliliter. The experiments were triplicated, and the standard errors of the mean are shown in Fig. 6.

**Statistical analysis.** Data points of all experiments represent means of the results from at least three independent experiments, and error bars indicate standard deviations. Significance of the differences was determined using the *t* test in Excel.

## ACKNOWLEDGMENTS

This work was supported by JSPS KAKENHI, Grant-in-Aid for Young Scientists, Grant Number 18K14378. We thank Takeshi Zendo for kindly providing purified lacticin Q and the support for LC-MS analyses. We thank Editage for English language editing.

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
