## [Reviewer comments · Microbiology Spectrum]

Microbiology Spectrum

Construction of leaderless-bacteriocin-producing bacteriophage targeting *E. coli* and neighboring Gram-positive pathogens

Yoshimitsu Masuda, Shun Kawabata, Tatsuya Uedoi, Ken-ichi Honjoh, and Takahisa Miyamoto

Corresponding Author(s): Yoshimitsu Masuda, Kyushu University

Review Timeline:

Submission Date:	April 29, 2021
Editorial Decision:	May 27, 2021
Revision Received:	June 13, 2021
Accepted:	June 23, 2021

Editor: Daria Van Tyne

Reviewer(s): The reviewers have opted to remain anonymous.

Transaction Report:

DOI: <https://doi.org/10.1128/Spectrum.00141-21>

May 27, 2021

Prof. Yoshimitsu Masuda
Kyushu University
Department of Bioscience and Biotechnology
Fukuoka
Japan

Re: Spectrum00141-21 (Construction of leaderless-bacteriocin-producing bacteriophage against *E. coli* and neighboring Gram-positive pathogens)

Dear Prof. Yoshimitsu Masuda:

Thank you for submitting your manuscript to Microbiology Spectrum. Your paper was reviewed by two expert reviewers, and their comments appear below. Both reviewers found merit to your study, however they also raise important issues that need to be addressed. Please modify your manuscript in line with the reviewers' feedback, ensuring that you address all of their comments. Additionally, in Figure 4 please make sure that white space appears between pictures that have been cropped and placed next to one another, as in panel B.

When submitting the revised version of your paper, please provide (1) point-by-point responses to the issues raised by the reviewers as file type "Response to Reviewers," not in your cover letter, and (2) a PDF file that indicates the changes from the original submission (by highlighting or underlining the changes) as file type "Marked Up Manuscript - For Review Only". Please use this link to submit your revised manuscript - we strongly recommend that you submit your paper within the next 60 days or reach out to me. Detailed information on submitting your revised paper are below.

Link Not Available

Sincerely,

Daria Van Tyne

Journals Department
Reviewer comments:

Reviewer #1 (Comments for the Author):

The manuscript by Masuda et al. presents a very interesting approach and concept for development of a novel type of combined antimicrobial with dual specificity, where the gene encoding a gram-positive bacteriocin is expressed from a Gram-negative phage. In the work, a leaderless bacteriocin is combined with T4 phage is shown as a proof of principle, but the concept can in theory be expanded to numerous other combinations. In addition to demonstrating the dual activity, the authors also assess the burst size and lytic activity of the modified phage compared to the parental phage. In general, the manuscript is clearly written and results nicely presented and discussed.

Some comments:

- The introduction could be expanded to provide more background information. Please describe more about the target range, biosynthesis and mechanism-of-action of lacticin Q in particular. For example, related to discussion (line 264-266); what is known about the activity of this bacteriocin when expressed in *E. coli*?
- Furthermore, some background about the T4 phage multiplication (including the dependency of *trxA*) and life cycle would be useful in the introduction.
- Regarding production and release of L_{nqQ}. The bacteriocin was not detected in the phage lysate and the authors discuss possible reasons for this. It would be interesting if the authors discuss possible strategies to increase the production of bacteriocin, eg. increasing the gene copy number, increasing the latency period.
- Along the same lines, the authors highlight the use of leaderless bacteriocins as ideal for the purpose, since they are non-modified and without a leader. However, if I interpret the results correctly, the leaderless bacteriocins are mostly released from the *E. coli* cells due to lysis and not export from intact cells. If this is true, I believe other non-modified bacteriocins could be equally well suited for the purpose (as long as they are expressed/cloned without their leader). Please discuss.
- Figure 3 and line 135-137. When the parental T7 phage was exposed to *Bacillus coagulans* overlay the authors state that "the growth of *B. coagulans* was shown inside the plaques of T7 phage formed against *E. coli*". I do appreciate that the plaques are not as clear as in the other panels and that there is growth on the plaques. However, there also seems to be a zone with no growth at the edges of the plaques. Could the authors explain or speculate on this?
- Figure 4A. The setup of the figure is a bit confusing due to the placement of the "Bc (Alone)" exposed to L_{nqQ} only. Consider reorganizing the figure to avoid confusion, e.g. by making a frame around this part of the figure.

Minor comments:

- Title: consider replacing "against" with "targeting"
- Line 20-21: remove the word "body"
- Line 25: insert the word "an" before "LLB-producing".

- Line 27: the *trxA*-gene needs some kind of introduction. Please rephrase to shortly explain the function of the *trxA* gene.
- Line 30: replace "plate" with "plates"
- Line 43: add "the" before "Gram-negative"
- Line 79: consider replacing "approach" with "classification". Remove the word "of"
- Line 80. To my knowledge, "listeria-specific bacteriocins" are not widely used a group in the classification of bacteriocins. Would be more suitable to use e.g., "most non-modified bacteriocins"
- Line 111: explain the expression dynamics of the T7 promoter
- Line 274: consider rephrasing to "... LLB phage could also slow down development of resistance to ...".
- Line 594: "10 10⁻⁸ PFU/ml". Please correct.
- References: formatting of bacterial names is lacking

Reviewer #2 (Comments for the Author):

The work describes the engineering of phage T7 to produce InqQ. This is an interesting approach and the engineering of phages in such a way can be useful for manipulating bacterial communities. In its current form the work seems preliminary. The engineering of phages generally is still in its infancy, so the production of an engineered phage is quite an achievement. Phage T7 has been previously engineered using *trxA* as a selective marker, so the method per se is not novel, only the gene added (which is interesting). The previous work that developed the approach of *trxA* should be cited.

Clearly phage mutants have been obtained and when used in a clever double-double agar overlay, there can be seen to be an inhibitory effect on *Bacillus*.

The amount of InqQ produced was not fully quantified and only tested on plates. If it inhibits growth when grown in liquid medium was not presented and would seem to be critical. If there is such tiny amounts, will the approach work if the bacteria are in liquid culture and not forced together as they are on plates ?

The data presented on one-step experiments is not conclusive. The text suggests there was a difference in burst size, but no statistical analysis was presented to support this. Given the small difference in burst size, it is not clear if the difference is simply a result of the variation that occurs when performing one-step experiments. To fully test if there are quantitative differences between the WT and mutant phage, the use of a phage virulence index (Storms 2020) would be more appropriate. This would incorporate all elements of phage replication

The explanation of the differences in observed phage production, is not supported by the data presented.

L20 - word missing in the sentence

L71 - clarify what is meant by stress ? Is this possibly better phrased as not stable in environmentally relevant conditions?

L73- probably needs a reference here

L105- Qimron work should be cited that showed this was essential

L177 are these reported differences statistically different ? Looking at the overlap of error bars in Figure 5, it is unclear if this is the case. The error associated with these numbers should be reported and tested for significant difference

L182- I am unclear on the rationale for this ? The authors have been thorough in testing this, but it is not clear why it would target a Gram-ve bacteria, when all published tests of this bacteriocin to date have shown it targets Gram +ve bacteria.

L189 - the change in the point when resistance is observed, is surprising. If InqQ does not effect growth of E.coli, why would it alter resistance ? This could be discussed further

L214 - as mentioned previously this method has previously been used and is not a novel technique for T7, previous work using this technique should be cited.

L216 - the method used for creating phage mutants in phage T7 is based on previously described methods: Qimron et al, Grigonyte et al and others have used for engineering phage T7 in the same manner. The fact this method has been used before should really be cited as the principle behind it is not new .

L230 - the differences between these two strains is intriguing. Also how accurate this increase is, is not clear. The level of production was not quantified between the two strains. Furthermore, the amount of phage produced appears to be the same between both phages based on the one step experiment. Certainly not a 20x difference in the number of phages produced. Why there would be a decrease in protein synthesis of just one protein (InqQ) and not all phage proteins is not clear. This suggests trxA has very specific role in the production of InqQ ? as well as its general role binding to the RNA polymerase subunits that allow transcription of T7.

L246 - all of the points raised around the use qPCR are true. But these have been raised previously with many other systems. eg qPCR has been used widely estimate burst size and latent period of cyanophages . Currently it reads as if this is the first time such an approach has been taken for phages, which it isnt , but might be for T7.

Figure 1 - it would help to have the gene numbers and accession on the region that is being targeted for recombination

What is driving the expression of the trxA ? is this a T7 promoter or Ecoli promoter .

Figure 6 - there are no error bars presented on this

Staff Comments:

Preparing Revision Guidelines

- Point-by-point responses to the issues raised by the reviewers in a file named "Response to Reviewers," NOT IN YOUR COVER LETTER.
- Upload a compare copy of the manuscript (without figures) as a "Marked-Up Manuscript" file.
- Each figure must be uploaded as a separate file, and any multipanel figures must be assembled into one file.
- Manuscript: A .DOC version of the revised manuscript
- Figures: Editable, high-resolution, individual figure files are required at revision, TIFF or EPS files are

preferred

For complete guidelines on revision requirements, please see the Instructions to Authors at [link to page]. **Submissions of a paper that does not conform to Microbiology Spectrum guidelines will delay acceptance of your manuscript.**

Please return the manuscript within 60 days; if you cannot complete the modification within this time period, please contact me. If you do not wish to modify the manuscript and prefer to submit it to another journal, please notify me of your decision immediately so that the manuscript may be formally withdrawn from consideration by Microbiology Spectrum.

If you would like to submit an image for consideration as the Featured Image for an issue, please contact Spectrum staff.

Response to reviewers

Answer for Reviewer 1

Thank you very much for reviewing our manuscript and giving us helpful comments. According to your kind suggestions, we revised our manuscript and revised parts are marked with yellow in the text.

1. Regarding the information about lacticin Q, as you suggested, we added more background information especially about mode of action and biosynthesis (line 96–104). So far, there have been no publication demonstrating directly lacticin Q expression in *E. coli* without any N-terminal extension, but from the result of previous study by Iwatani et al. (ref. 20), it is assumed that lacticin Q can be immediately active after translation in *E. coli* as well. In addition, it is not published data but when we tried to introduce expression plasmid with lacticin Q structure gene with T7 promoter into *E. coli* BL21 carrying T7 RNA polymerase, transformation could not succeed. Therefore, we used DH5 α strain for homologous recombination to avoid leaked lacticin Q expression which damage *E. coli*.
2. As you recommended, we added more information about T7 phage, especially about its genome replication and the role of thioredoxin in it (line 106–110).
3. Regarding the production and release of LncQ, we discussed shortly at line 255–257, and we also added another sentence at line 296–298.
4. As you suggested it is theoretically possible that not only LLB but other mature peptides from other non-modified bacteriocins will be used as candidates. So, we described this in discussion part at line 299–301.
5. Regarding the clear area between *E. coli* growth and *B. coagulans* growth in T7 phage halo (Fig. 3), we are sorry, but we have no concrete explanation based on any experiments, but we assume that this could be because of some compounds produced from *E. coli* and defused into halo area which inhibited growth of *B. coagulans*. Because this is not actually related with our purpose in this study, we did not mention in our manuscript.
6. Regarding Fig. 4, we revised its setup as you suggested.

Response to minor comments

Title: we replaced “against” with “targeting” .

Line 20–21: we removed “body” .

Line 25: we revised as you suggested.

Line 27: we added short explanation of *trxA*.

Line 30: we replaced “plate” with “plates” .

Line 44: we added “the” .

Line 80: we replaced “approach” with “classification” .

Line 81: we replaced “listeria-specific bacteriocins” with “non-modified bacteriocins” .

Line 111: we added explanation of T7 phage gene expression including T7 promoter at line

127–132.

Line 274: we revised as you suggested at line 291.

Line 594: we corrected as “ 10^{8-10} ” at line 646.

References: we corrected bacterial names.

Answer for Reviewer 2

Thank you very much for reviewing our manuscript and giving us helpful comments. According to your kind suggestions, we revised our manuscript and revised parts are marked with yellow in the text.

1. Regarding previous reports about *trxA* by Qimron *et al.*, we are very sorry that we forgot to add references, and as you suggested, we added these citations at proper places in revised manuscript.
2. Regarding the challenge for tiny productivity of *InqQ*, as you suggested, this must be improved in future by such as 1) re-evaluating the position of LLB structure gene for suitable expression timing or 2) increasing the number of structure gene to enhance expression level. We discussed this shortly at line 255–257, and we also added another sentence at line 296–298
3. Regarding the one-step growth experiments, we used constructed sigmoidal curves for determining burst sizes but as you suggested, experimental raw data are more suitable to evaluate statistical differences. Therefore, we revised the part of one-step growth experiments at line 192–204. In revised manuscript, the significance of difference of burst size was evaluated by T-test and p value was 0.1329. The infectious times for both phages were also described more clearly. The use of a phage virulence index by Storms is also good protocol to evaluate phage activity, but unfortunately, we do not have the plate reader machine which allow us to perform continuous bacterial culture experiments, and our protocol is also appropriate enough to evaluate the difference between T7 and *InqQ*-T7 phages.

Response to minor comments

Line 20: we are sorry, but we cannot figure out what word is missing in this sentence.

Could

you tell us more precisely?

Line 71: we revised as “not stable in environmentally relevant conditions” .

Line 73: we combined two sentence and added reference (7) at line 73–77.

Line 105: we added the information of *trxA* and citations of Qimron’ s works at line 106–110.

Line 177: we revised sentences and added statistical analysis for one-step growth experiments

at line 192–204 as mentioned above Answer No. 3.

Line 182: Regarding the activity of lactacin Q inside *E. coli*, to support the

understanding, we

added more background information about lactacin Q especially its mode of action

and biosynthesis at line 96–104. So far, there have been no publication demonstrating directly lactacin Q expression in *E. coli* without any N-terminal

extension, but from the result of previous study by Iwatani et al. (ref. 20), it is

assumed that lactacin Q can be immediately active after translation in *E. coli* as well.

In addition, it is not published data but when we tried to introduce expression plasmid carrying lactacin Q structure gene with T7 promoter into *E. coli* BL21 carrying T7 RNA polymerase, transformation could not succeed. Therefore, we used

DH5 α strain for homologous recombination to avoid leaked lactacin Q expression

which damage *E. coli*.

Line 189: As described at line 100–103 in revised manuscript, lactacin Q does not require any

specific target molecule to attack negatively charged cell membrane, which suggest

the lactacin Q effect from inside *E. coli* cell.

Line 214, 216: we added references 24 and 25 at line 241 and 347 as you suggested.

Line 230: as you mentioned, this differences between two host strains are of interest for us as

well. However, unfortunately, we have no experimental or published information to

explain specific impact of *trxA* on lactacin Q production. We hypothesized reduced

protein synthesis in general as one of the reasons. Therefore, to avoid this *trxA*

dependent problem, we are trying to establish new platforms with CRISPR–Cas systems in which we do not need to manipulate host strains genomes.

Line 246: we are very sorry that we did not realize the previous reports precisely demonstrating qPCR method to quantify phages. Therefore, we deleted the discussion part mentioning qPCR method, and we added reference 27 by Peng *et al.*

at line 189 and 437.

Figure 1: we added gene numbers such as gp10 and gp11. *trxA* and *lnqQ* are under T7

promoter, so they will be expressed by T7 RNA polymerase.

Figure 6: there were error bars but hidden behind the symbols, so we shrank the size of

symbols in revised Figure 6.

June 23, 2021

Prof. Yoshimitsu Masuda
Kyushu University
Department of Bioscience and Biotechnology
Fukuoka
Japan

Re: Spectrum00141-21R1 (Construction of leaderless-bacteriocin-producing bacteriophage targeting *E. coli* and neighboring Gram-positive pathogens)

Dear Prof. Yoshimitsu Masuda:

Thank you for submitting a revised version of your paper. Your manuscript has now been accepted, and I am forwarding it to the ASM Journals Department for publication. You will be notified when your proofs are ready to be viewed.

Sincerely,

Daria Van Tyne
Editor, Microbiology Spectrum
